Different prevalence and spectrum of malignancy between Chinese patients and American patients with rheumatoid arthritis: a comparative study

Ouyang Zhi-Ming 1
Zou Yao-Wei 1
Pan Jie 1
Lu Ye 1
Yang Ying 1
Li Qian-Hua 1 2
Ma Jian-Da 1
Jia Pei-Wen 1
Wu Tao 1
Fan Yu-Ting 1
http://orcid.org/0000-0002-0973-7063 Lin Jian-Zi 1
Wei Xiu-Ning 1
Yang Kui-Min 1
Su Yun 3
Dai Lie 1 dailie@mail.sysu.edu.cn
1 Department of Rheumatology and Immunology, Sun Yat-sen Memorial Hospital, Sun Yat-sen University , Guangzhou, Guangdong , China
2 Department of Rheumatology and Immunology, Shenshan Medical Center, Memorial Hospital of Sun Yat-sen University , Shanwei, Guangdong , China
3 Zhongshan School of Medicine, Sun Yat-sen University , Guangzhou, Guangdong , China
Zhang Xin
Electronic publication date: 2024 Dec 18
Publication date: 2024
Volume: 12
Electronic Location ID: e18650
Received 2024 Jun 17; Accepted 2024 Nov 15
Copyright: © 2024 Ouyang et al.
Copyright year: 2024
Copyright holder: Ouyang et al.
License: This is an open access article distributed under the terms of the Creative Commons Attribution License, which permits unrestricted use, distribution, reproduction and adaptation in any medium and for any purpose provided that it is properly attributed. For attribution, the original author(s), title, publication source (PeerJ) and either DOI or URL of the article must be cited.
License URL: https://creativecommons.org/licenses/by/4.0/

Keywords: Rheumatoid arthritis, Malignancy, Chinese, American

Funding: Chinese National Key Technology R&D Program, Ministry of Science and Technology 2022YFC2504600 and 2022YFC2504601 National Natural Science Foundation of China 82171780, 81971527, and 82101892 Guangdong Basic and Applied Basic Research Foundation 2022A1515010524 and 2023A1515030253 This work was supported by the Chinese National Key Technology R&D Program, Ministry of Science and Technology (2022YFC2504600 and 2022YFC2504601), the National Natural Science Foundation of China (82171780, 81971527, and 82101892), and the Guangdong Basic and Applied Basic Research Foundation (2022A1515010524 and 2023A1515030253). The funders had no role in study design, data collection and analysis, decision to publish, or preparation of the manuscript.

==============================
Objective

To characterize the epidemiological characteristics of malignancy in Chinese patients with rheumatoid arthritis (RA) versus American patients and investigate their associated factors.

Methods

Data were collected from a real-world Chinese RA population and American patients with RA from the National Health and Nutritional Examination Survey. The prevalence and subtypes of malignancy and their potential associated factors were investigated in both populations.

Results

A total of 2,073 Chinese and 2,928 American patients with RA were included. There was a lower prevalence of malignancy in Chinese than in their American counterparts before (5.7% vs. 17.1%) and after matching (6.2% vs. 12.6%, both P < 0.001). Gender discrepancies in malignancy prevalence were observed, with a male predilection for RA with malignancy in China (8.2% vs. 5.5%), while it was the opposite in American patients (10.1% vs. 13.5%, both P < 0.05). The top type of malignancy among male patients with RA was lung cancer in Chinese (2.29%), but non-melanoma skin cancer (3.43%) in American; while among female patients was breast cancer both in Chinese (1.72%) and American (3.43%). Multivariate logistic regression analyses showed that older age (odds ratio (OR) = 1.050) and positive anti-cyclic citrullinated peptide antibody (OR = 2.752) were independently associated with malignancy in Chinese patients with RA, while female (OR = 1.395), older age (OR = 1.033), active smoking (OR = 1.580) and cardiovascular diseases (OR = 1.523) in American patients.

Conclusion

The prevalence, subtypes and risk factors of malignancy were substantially different in Chinese patients with RA and their American counterparts, which implied the importance of individualized malignancy screening strategies for patients with RA.

Introduction

Rheumatoid arthritis (RA) is a condition characterized by chronic inflammation associated with progressive joint damage, disability, and deterioration of quality of life (Di Matteo, Bathon & Emery, 2023). Comorbidities are highly prevalent in RA, in which malignancy has raised critical attention in the long-term management of patients with RA (Figus et al., 2021; De Cock & Hyrich, 2018; Sonomoto & Tanaka, 2023). Malignancy is a major underlying cause of death in RA, together with respiratory diseases and cardiovascular diseases (Black et al., 2023; Lee et al., 2022). Published studies have uncovered an increased risk of overall malignancy in patients with RA compared with the general population, with a standardized incidence ratio (SIR) ranging from 1.09 to 3.99 (Simon et al., 2015; Beydon et al., 2023; Wang et al., 2023). In a large-scale retrospective Chinese cohort, patients with RA also demonstrated an increased risk of developing malignancy (SIR 3.99, 95% confidence interval (CI) [3.40–4.65]) (Zhou et al., 2022). The malignancy burden raises the awareness of reporting, screening and prevention in the daily practice of RA management.

According to recently published data, the cancer incidence rate as well as the cancer spectrum in China differ from that in Western developed countries (Qiu, Cao & Xu, 2021; Xia et al., 2022; Ju et al., 2023). The age-SIR of all cancers in China (204.80 per 100,000 population) was comparable to that in the world (201.00 per 100,000 population) but lower than those in the United States (US, 362.20 per 100,000 population) and the United Kingdom (UK, 319.90 per 100,000 population) based on the global epidemiological data released in 2020 (GLOBOCAN 2020) (Qiu, Cao & Xu, 2021). Unlike in the US and the UK, where prostate cancer (male) and breast cancer (female) were the mainstay, lung cancer dominated in China (both sexes). Liver, stomach, esophageal, and cervical cancer were more commonly seen in China than in the US and UK (Qiu, Cao & Xu, 2021). Meanwhile, studies reported that Asian patients with RA demonstrated fewer malignancies (0.6–5.0%), while other studies on RA patients in Western countries reported more patients with a history of malignancy (8.3–12.5%) (Dougados et al., 2014; Jin et al., 2015, 2017; Choi et al., 2018). However, there is a lack of comparative studies between patients with RA in China and Western countries concerning malignancy burden and their spectrum. The diversity of malignancy types by different regions should be taken into consideration in the management of RA. Given the availability of data from a nationally representative sample in the US National Health and Nutritional Examination Survey (NHANES) to the public, we aimed to characterize the prevalence of overall malignancy and major site-specific cancers in Chinese RA patients relative to the American RA patients from the NHANES, and to explore the risk factors of malignancy in the subgroup analysis of Chinese patients with RA and their US counterparts, respectively.

Methods

Study design, data sources, and study populations

A 1:1 individually matched case-case comparative study was conducted to characterize malignancy epidemiological characteristics of patients with RA between China and the US, where cases were patients with RA. The matching was described in the “statistical analysis” section.

Chinese patients with RA

Chinese patients with RA were consecutively recruited by rheumatologists from January 2001 to January 2024 at the Department of Rheumatology and Immunology, Sun Yat-sen Memorial Hospital, and also recruited from August 2022 to January 2024 at the Department of Rheumatology and Immunology, Shenshan Medical Center, Memorial Hospital of Sun Yat-sen University, China (Lin et al., 2019). The patients with RA were clinically diagnosed, and their diagnoses also fulfilled the 1987 American College of Rheumatology (ACR) (Arnett et al., 1988) or 2010 ACR/European League Against Rheumatism (EULAR) classification criteria (Aletaha et al., 2010). Patients ≥18 years old, with data available on demographics and history of comorbidity at enrollment were included in this study. Those who overlapped with other autoimmune diseases were excluded. The Ethics Committee of Sun Yat-sen Memorial Hospital approved this study (SYSEC-2009-06). All patients consented to participate and signed informed consent.

The US patients with RA

The US patients with RA were selected from the US NHANES database 1999–2018. The NHANES has become a continuous annual survey of a national sample of approximately 5,000 persons representative of the noninstitutional civilian resident population in the US since 1999 (Zipf et al., 2013). The NHANES program was performed by the Centers for Disease Control and Prevention of the US. A multistage, probability sampling method with oversampling of several subgroups was used to collect data with in-person face-to-face interviews and physical examinations (Zipf et al., 2013). Participants came from diverse racial and ethnic groups, including non-Hispanic Whites, non-Hispanic Blacks, other Hispanic Americans, Mexican Americans, and other racial/ethnic Americans. Data were released to the public in 2-year cycles with a 2-year sample size of approximately 10,000 individuals (Centers for Disease Control and Prevention (CDC). National Center for Health Statistics (NCHS), 2023). We used data from 10 data-releasing cycles covering a period of 1999–2018. Arthritis in the NHANES was classified into RA, osteoarthritis, and other types of arthritis. Participants with arthritis were identified if they answered “Yes” to the question “Doctor ever said you had arthritis?” in the medical condition questionnaire. If their arthritis was classified as RA (Liu et al., 2023; Liang et al., 2022), they were deemed patients with RA. In this study, we included American patients with RA with data available on demographics and history of comorbidity. Those who overlapped with other autoimmune diseases were excluded.

Demographic and clinical data

Demographic and clinical data of Chinese patients with RA were collected at enrollment as we previously reported (Lin et al., 2019; Pan et al., 2022), including gender, age, active smoking, body mass index (BMI) and clinical data of RA (RA disease duration, erythrocyte sedimentation rate (ESR), C-reactive protein (CRP), rheumatoid factor (RF) status, anti-cyclic citrullinated peptide antibody (ACPA) status, RA disease activity, physical function and comorbidity). Disease activity was assessed by the clinical disease activity Index (CDAI). Physical dysfunction was assessed by the Stanford Health Assessment Questionnaire disability index (HAQ-DI).

Information of US patients with RA available in NHANES and equivalent to Chinese patients was selected, such as gender, age, active smoking, BMI, race/ethnicity, CRP, and comorbidity (Liu et al., 2023; Liang et al., 2022).

History of malignancy

In Chinese patients with RA, a history of malignancy was collected by face-to-face questionnaire survey combined with verification through medical records. Malignancy diagnosis was confirmed by histopathology or postoperative pathology according to the International Classification of Diseases (ICD-10) criteria (World Health Organization, 2016).

In the US patients with RA, a history of malignancy is available in NHANES, and the medical conditions section provides self-reported health condition information (Cao, Friedenreich & Yang, 2022). Malignancy diagnosis was based on the following two questions: 1. “Have you ever been told by a doctor or other health professional that you had cancer or a malignancy of any kind?”; 2. “What kind of cancer was it and when it was diagnosed?”. These questions were asked, at home, by trained interviewers using the Computer-Assisted Personal Interviewing (CAPI) system. The CAPI system is programmed with built-in consistency checks to reduce data entry errors.

Statistical analysis

Chinese patients with RA were initially stratified by gender, followed by matching individual patient within each gender stratum based on age with one American patient. This two-step matching strategy aimed to balance the heterogeneity of age and gender across Chinese and US patients with RA.

In the Chinese RA population, the mean ± standard deviation (SD) was used for normally distributed continuous variables and median (interquartile limits (IQL) 25th, 75th centiles) for non-normally distributed continuous variables, while categorical variables were expressed as the number (percentage, %). Sampling weights were used in accordance with the NHANES analytic guidelines to account for its complex multistage sampling designs and to calculate nationally representative estimates (Johnson et al., 2013). The weights are equal to two-tenths of the weights for 1999–2002 or one-tenth of the weights for 2003–2018 (Johnson et al., 2013). A weighted mean and its 95% CI as well as a weighted median and its IQL were calculated for a continuous variable, while an unweighted number, its relevant weighted percentage and the 95% CI of the weighted percentage were calculated for a categorical variable.

We used the weighted t-tests or Mann–Whitney U test to analyze between-group differences for continuous variables, and the weighted Chi-squared tests or Fisher’s exact tests for categorical variables, while the weights in Chinese patients with RA were set to 1 due to RA patients in China in this study were collected from two centers using a simple sampling method. A (weighted) multiple logistic regression model was composed to identify significant predictors for malignancy. SPSS (IBM Corp. released 2017; IBM SPSS Statistics for Windows, Version 25.0; IBM Corp., Armonk, NY, USA) and R (version 4.2.2) software were utilized for all analyses. A P value ≤ 0.05 from the two-sided test was considered statistically significant.

Results

Demographic and clinical characteristics of Chinese and US patients with RA

A total of 2,073 Chinese patients with RA were included in this study after 68 patients were excluded from 2,141 Chinese patients with RA in the parent study because of overlap with other autoimmune diseases (Fig. S1). The mean age of Chinese patients with RA was 53.6 with a SD of 12.9 years, and 1,629 (78.6%) were female. The median disease duration was 45 (IQL 12-114) months. There were 1,744 (84.1%) patients with active disease (CDAI > 2.8, Table 1).

Table 1 Comparisons of characteristics between Chinese patients and US patients with RA before and after matching.

Characteristics	Before matched	After matched	
Chinese RA (n = 2,073)	US RA (n = 2,928)	P	Chinese RA (n = 1,660)	US RA (n = 1,660)	P	
Female*, n (%)	1,629 (78.6)	1,752 (59.4) [56.8– 61.9]	<0.001	1,223 (73.7)	1,223 (73.7) [69.2–75.5]	1.00	
Age*, years	53.6 ± 12.9	57.9 (57.1, 58.6)	<0.001	55.9 ± 12.3	55.8 (52.0, 63.7)	0.86	
Active smoking1, n (%)	334 (16.1)	1,614 (58.2) [55.3–61.0]	<0.001	321 (19.3)	900 (59.5) [55.7–63.2]	<0.001	
RA disease duration, months	45 (12, 114)	–	–	–	–	–	
BMI2, kg/m2	21.9 ± 3.3	29.2 (25.0, 34.1)	<0.001	22.0 ± 3.2	29.6 (24.9, 34.9)	<0.001	
BMI categories, n (%)			<0.001			<0.001	
Underweight	324 (15.6)	32 (1.2) [0.7–1.9]		251 (15.1)	20 (1.4) [0.8–2.5]		
Normal weight	1,226 (59.1)	595 (23.6) [21.4–25.9]		981 (59.1)	321 (23.7) [20.9–26.8]		
Overweight	435 (21.0)	823 (29.4) [27.3–31.5]		362 (21.8)	435 (27.4) [24.6–30.4]		
Obese	88 (4.2)	1,264 (45.9) [43.3–48.5]		66 (4.0)	778 (47.5) [44.0–51.1]		
CRP3, mg/L	6.0 (3.3, 23.0)	5.1 (1.9, 15.7)	<0.001	6.6 (3.3, 25.3)	4.0 (1.7, 10.8)	<0.001	
Comorbidity, n (%)							
Malignancy	118 (5.7)	449 (16.2) [14.4–18.3]	<0.001	103 (6.2)	209 (14.5) [12.2–17.2]	<0.001	
Hypertension4	593 (28.6)	1,960 (61.2) [58.1–64.2]	<0.001	482 (29.0)	999 (55.9) [52.0–59.7]	<0.001	
Diabetes5	253 (12.2)	902 (24.0) [22.1–26.4]	<0.001	202 (12.2)	446 (20.9) [18.4–23.6]	<0.001	
Dyslipidemia7	582 (28.1)	2,178 (78.5) [76.3–80.7]	<0.001	459 (27.7)	1,225 (77.7) [74.8–80.4]	<0.001	
Cardiovascular diseases6	194 (9.4)	803 (24.2) [22.1–26.4]	<0.001	156 (9.4)	370 (20.4) [18.0–23.0]	<0.001	
Previous medications							
Glucocorticoids, n (%)	1,029 (49.6)	201 (7.6) [5.8–9.7]	<0.001	815 (49.1)	119 (7.4) [5.6–9.5]	<0.001	
csDMARDs, n (%)	1,455 (70.2)	218 (8.7) [6.6–10.9]	<0.001	1,160 (69.9)	122 (8.6) [6.3–10.8]	<0.001	
bDMADRs, n (%)	142 (6.8)	51 (2.0) [0.9–3.6]	<0.001	115 (6.9)	28 (2.0) [1.0–3.5]	<0.001	
tsDMARDs, n (%)	71 (3.4)	1 (0.1) [0–0.2]	<0.001	58 (3.5)	1 (0.1) [0–0.2]	<0.001	
Notes:

* Used for matching.

1 Active smoking was only available in 2,924 and 1,658 US RA patients before and after matching.

2 BMI was only available in 2,710 and 1,554 US RA patients before and after matching.

3 CRP was only available in 2,147 and 1,313 US RA patients before and after matching, and the normal range for CRP was less than 5.0 mg/L.

4 Hypertension was only available in 2,926 and 1,659 US RA patients before and after matching.

5 Diabetes was only available in 2,914 and 1,646 US RA patients before and after matching.

6 Cardiovascular diseases were only available in 2,927 and 1,660 US RA patients before and after matching.

7 Dyslipidemia was only available in 2,798 and 1,592 US RA patients before and after matching.

Data in Chinese RA patients are shown as number (percentage, %), mean ± SD, and median (IQL, 25th, 75th) if appropriate.

Data in US RA patients are shown as unweighted number (weighted percentage, %) [95% CI], weighted mean [95% CI] and weighted median (IQL, 25th, 75th) if appropriate.

RA, rheumatoid arthritis; BMI, body mass index; CRP, C-reactive protein; SD, standard deviation; IQL, interquartile limits. csDMARDs, conventional synthetic disease-modifying antirheumatic drugs; bDMADRs, biological disease-modifying antirheumatic drugs; tsDMARDs, targeted Synthetic Disease-Modifying Anti-Rheumatic Drugs.

Among the 2,952 US patients with RA from NHANES, 20 patients overlapping with other autoimmune diseases and four patients missing data of malignancy were excluded using the same criteria as the Chinese patients (Fig. S1). The mean age of the included 2,928 US patients with RA was 58.0 (95% CI [57.1–58.8]) years, and 1,749 (59.8%) were female.

There were more female patients in Chinese RA while the US patients with RA were older. In addition, Chinese patients with RA had a significantly higher level of CRP but lower level of BMI, as well as a lower prevalence of active smoking, hypertension, diabetes, dyslipidemia and cardiovascular diseases than their US counterparts. Furthermore, Chinese patients with RA were more likely to have previous uses of glucocorticoids (49.6% vs. 7.6%), conventional synthetic disease-modifying anti-rheumatic drugs (csDMARDs, 70.2% vs. 8.7%), biologic DMARDs (bDMARDs, 6.8% vs. 2.0%), and targeted synthetic DMARDs (tsDMARDs, 3.4% vs. 0.1%, all P < 0.05, Table 1).

Prevalence of malignancy in Chinese patients and matched US patients with RA among different stratification

A total of 118 (5.7%) Chinese patients with RA had malignancy, which was significantly lower than that in the US patients with RA (17.1%, P < 0.001, Table 1). An individual matching was conducted to balance the effects of age and gender on the prevalence of malignancy between two populations, and a total of 1,660 Chinese patients with RA were age- and gender-matched to the US patients with RA individually at a ratio of 1 to 1 (Fig. S1). Chinese patients with RA still had a lower prevalence of malignancy than matched US patients with RA (6.2% vs. 12.6%). Surprisingly, gender discrepancies in malignancy prevalence were observed between Chinese and US patients with RA, in which the prevalence in male patients was higher than that of female patients with RA in China (8.2% vs. 5.5%), while the prevalence in male patients was lower than that in female patients with RA in the US (10.1% vs. 13.5%). In addition, the prevalence of malignancy in female patients with RA in the the US was higher than that in China (13.5% vs. 5.5%), while there was no significant difference in male patients. The prevalence of malignancy was lower in RA patients in China than those in the US both in patients with or without smoking (with smoking: 9.3% vs. 14.8%; without smoking: 5.5% vs. 10.0%, all P < 0.05, Fig. 1A).

Figure 1 The prevalence of malignancy in Chinese patients and matched US patients with RA.

The prevalence of malignancy in all, male, female, with and without active smoking Chinese patients and matched US patients with RA (A); The prevalence of malignancy among age subgroup in all (B), male (C) and female (D) Chinese patients and matched US patients with RA. *P < 0.05; **P < 0.01; ***P < 0.001, Comparisons of prevalence of malignancy among age subgroup in China and US patients with RA in figure B, C and D.

Further age stratification showed the prevalence of malignancy greatly increased after 30 years old with 1.3%, 4.7%, 6.0%, 7.4%, and 12.6% in Chinese patients with RA of 31–40, 41–50, 51–60, 61–70, ≥71 years old, respectively (Fig. 1B). However, the prevalence of malignancy only greatly increased after 70 years old in US patients with RA (≤70 years old: 10.0–12.9%, ≥71 years old: 20.0%, Fig. 1B). Further gender stratification showed both male and female patients had similar trends with a total population in Chinese patients with RA, while there was no significant difference among age subgroups in female US patients with RA. In addition, the prevalence of malignancy in the US patients with RA was higher than in China among age subgroups, but there was no significant difference in male (Figs. 1B–1D).

Profile of malignancy types in Chinese patients and matched US patients with RA

Further analysis of malignancy types found that the highest prevalence of malignancy in Chinese patients with RA was breast cancer (1.27%), followed by lung cancer (1.08%), colon cancer (0.84%), thyroid cancer (0.66%) and bladder cancer (0.36%, Fig. 2A). Differently, the top five types of malignancy among the US patients with RA were breast cancer (2.53%), cervix cancer (1.99%), non-melanoma skin cancer (1.87%), uterus cancer (1.14%) and colon cancer (0.72%). There was 0.24% lymphoma in the US patients with RA, while there was no lymphoma in the Chinese cohort together with only 0.18% non-melanoma skin cancer (Fig. 2A).

Figure 2 The prevalence of malignancy types in Chinese patients and matched US patients with RA.

The prevalence of malignancy types in all (A), female (B) and male (C) Chinese patients and matched US patients with RA.

Further gender stratification showed that the highest prevalence of malignancy in Chinese female patients with RA was breast cancer (1.72%), followed by colon cancer (0.74%), thyroid cancer (0.74%), lung cancer (0.65%) and bladder cancer (0.33%, Fig. 2B). Differently, the top five types of malignancy among US female patients with RA were breast cancer (3.43%), cervix cancer (2.70%), uterus cancer (1.55%), non-melanoma skin cancer (1.31%) and ovary cancer (0.90%, Fig. 2B). While, the top type of malignancy among male patients with RA was lung cancer in Chinese (2.29%), followed by lung cancer (1.14%), colon cancer (0.92%), liver cancer (0.92%), nasopharynx cancer (0.69%) and esophagus cancer (0.69%, Fig. 2C). For male patients with RA in the US, the top five types of malignancy were non-melanoma skin cancer (3.43%), prostate cancer (2.52%), colon cancer (0.92%), testis cancer (0.46%) and lymphoma (0.23%, Fig. 2C).

The time interval between RA onset and malignancy diagnosis in Chinese patients with RA

Among 118 Chinese patients with RA and malignancy, the mean age of RA onset was 52.6 ± 12.8 years and their peak RA-onset age was 49–51 years. Their mean age of malignancy diagnosis was 55.3 ± 11.6 years and the peak age of malignancy diagnosis was 52–54 years, which was late of RA onset age (Fig. 3A). There were 75 (63.6%) patients with malignancy diagnosed after RA onset, and approximately 24.6% of the patients diagnosed with malignancy within the first 3 years after the onset of RA (Fig. 3B).

Figure 3 Time interval between the RA onset and malignancy diagnosis in Chinese patients with RA.

The distribution of the age of RA onset and malignancy diagnosis in Chinese patients with RA and malignancy (A); the distribution of time interval between the age of RA onset and malignancy diagnosis (B).

In NHANES, most RA patients in the US missed the diagnosis age of malignancy, except for 14 patients having a clear diagnosis age of malignancy.

Demographic and clinical characteristics in Chinese and US patients with RA and malignancy

In both Chinese and US patients with RA, patients with malignancy were significantly older than those without malignancy (Chinese: 60.2 years vs. 53.2 years; US: 64.1 years vs. 56.7 years, both P < 0.05, Table 2). However, among Chinese patients with RA, compared with those without malignancy, patients with malignancy were more male (30.5% vs. 20.9%), had a higher proportion of positive RF (86.4% vs. 69.4%) and positive ACPA (88.1% vs. 69.3%), as well as a higher level of CRP (median 12.1 mg/L vs. 6.4 mg/L), and were more likely to have previous uses of glucocorticoids (61.9% vs. 48.9%) and csDMARDs (83.1% vs. 69.4%), but less uses of bDMARDs (1.7% vs. 7.2%, all P < 0.05). While, among the US patients with RA, compared with those without malignancy, patients with malignancy were more female (67.7% vs. 57.8%), had a higher proportion of active smoking (61.8% vs. 57.5%), dyslipidemia (83.9% vs. 77.5%) and cardiovascular diseases (34.8% vs. 22.1%, all P < 0.05), but no difference in previous medication treatment.

Table 2 The clinical characteristics of Chinese patients and the US patients with RA and malignancy.

Characteristics	Chinese RA patients	US RA patients	
Without malignancy (n = 1,955)	With malignancy (n = 118)	P	Without malignancy (n = 2,479)	With malignancy (n = 449)	P	
Female, n (%)	1,547 (79.1)	82 (69.5)	0.01	1,473 (57.8) [54.9–60.5]	279 (67.7) [61.4–73.3]	0.03	
Age, years	53.2 ± 13.0	60.2 ± 10.4	<0.001	56.7 (55.9, 57.4)	64.1 (62.5, 65.7)	0.01	
RA disease duration, months	45 (12, 111)	56 (19, 129)	0.04	–	–	–	
Active smoking, n (%)	303 (15.5)	31 (26.3)	0.002	1,334 (57.5) [54.4–60.5]	280 (61.8) [55.9–67.3]	0.004	
BMI2, kg/m2	21.9 ± 3.3	22.2 ± 3.3	0.36	29.4 (25.0, 34.4)	28.6 (25.3, 33.4)	0.89	
BMI categories, n (%)			0.61			0.37	
Underweight	307 (15.7)	17 (14.4)		28 (1.1) [0.7–1.7]	4 (1.7) [0.4–6.3]		
Normal weight	1,156 (59.1)	70 (59.3)		505 (23.9) [21.4–26.7]	90 (21.6) [17.3–26.7]		
Overweight	410 (21.0)	25 (21.2)		685 (28.6) [26.3–30.9]	138 (33.4) [28.5–38.6]		
Obese	82 (4.2)	6 (5.1)		1,087 (46.4) [43.5–49.3]	177 (43.3) [37.1–49.7]		
Positive RF, n (%)	1,357 (69.4)	102 (86.4)	<0.001	–	–	–	
Positive ACPA, n (%)	1,354 (69.3)	104 (88.1)	<0.001	–	–	–	
Disease activity indicators							
28TJC	3 (1, 9)	5 (1, 8)	0.34	–	–	–	
28SJC	2 (0, 6)	2 (1, 6)	0.12	–	–	–	
PtGA	4 (2, 6)	3 (1, 5)	0.24	–	–	–	
PrGA	3 (2, 6)	3 (1, 5)	0.20	–	–	–	
Pain VAS	3 (2, 5)	3 (1, 5)	0.31	–	–	–	
ESR, mm/h	37 (20, 68)	46 (21, 79)	0.13	–	–	–	
CRP, mg/L	6.4 (3.3, 24.5)	12.1 (3.6, 39.0)	0.01	5.2 (1.8, 16.0)	4.3 (2.3, 10.6)	0.22	
CDAI	14 (6, 25)	15 (8, 22)	0.78	–	–	–	
HAQ-DI	0.4 (0, 1.0)	0.5 (0.3, 1.1)	0.28	–	–	–	
Comorbidity, n (%)							
Hypertension	559 (28.6)	34 (28.8)	0.96	1,639 (60.0) [56.7–63.1]	321 (67.5) [61.6–72.9]	0.44	
Diabetes	236 (12.1)	17 (14.4)	0.45	760 (23.8) [21.7–26.0]	142 (25.4) [20.6–30.8]	0.20	
Dyslipidemia	548 (28.0)	34 (28.8)	0.85	1,836 (77.5) [75.0–79.8]	342 (83.9) [79.1–87.7]	0.02	
Cardiovascular diseases	183 (9.4)	11 (9.3)	0.99	630 (22.1) [20.0–24.4]	173 (34.8) [29.1–41.0]	<0.001	
Previous medications							
Glucocorticoids, n (%)	956 (48.9)	73 (61.9)	0.006	170 (7.6) [5.8–9.7]	31 (7.4) [5.6–9.4]	0.84	
csDMARDs, n (%)	1,357 (69.4)	98 (83.1)	0.002	181 (8.7) [6.7–10.8]	37 (9.0) [7.1–11.0]	0.86	
bDMADRs, n (%)	140 (7.2)	2 (1.7)	0.02	44 (2.1) [1.1–3.2]	7 (1.7) [0.9–2.9]	0.59	
tsDMARDs, n (%)	70 (3.6)	1 (0.8)	0.18	1 (0.1) [0–0.2]	0 (0.0) [0–0]	1.00	
Notes:

Data in Chinese RA patients are shown as number (percentage, %), mean ± SD, and median (IQL, 25th, 75th) if appropriate; Data in US RA patients are shown as unweighted number (weighted percentage, %) [95% CI], weighted mean (95% CI) and weighted median (IQL, 25th, 75th) if appropriate.

RA, rheumatoid arthritis; RF, rheumatoid factor; ACPA, anti-cyclic citrullinated peptide antibody; 28TJC, 28-joint tender joint counts; 28SJC, 28-joint swollen joint counts; PtGA, patient global assessment of disease activity; PrGA, provider global assessment of disease activity; Pain VAS, pain visual analog scale; ESR, erythrocyte sedimentation rate; CRP, C-reactive protein; CDAI, Clinical Disease Activity Index; HAQ-DI, health assessment questionnaire disability index; csDMARDs, conventional synthetic disease-modifying antirheumatic drugs; bDMADRs, biological disease-modifying antirheumatic drugs; tsDMARDs, targeted Synthetic Disease-Modifying Anti-Rheumatic Drugs.

Further comparison in Chinese patients with malignancy after RA onset, we also found that patients with malignancy after RA onset had longer RA disease duration, more severe RA, including higher rate of positive rheumatoid factor (RF), as well as higher levels of patient global assessment of disease activity (PtGA), provider global assesment of disease activity (PrGA), pain visual analog scale (VAS), ESR, CDAI and HAQ-DI (Table S1). These data imply the potential relationship of malignancy with RA.

Logistical regression analysis of the associated factors with malignancy in Chinese patients and the US patients with RA

To explore the potential associated factors of malignancy in patients with RA, multivariate logistic regression analyses were performed (Table 3). In separate logistic regression analyses for individuals in Chinese and US, the result showed that older age (odds ratio (OR) = 1.050, 95% CI [1.032–1.068]) and positive ACPA (OR = 2.752, 95% CI [1.492–5.076]) were independently associated with malignancy in Chinese patients with RA. Further gender stratification showed that older age (OR = 1.070, 95% CI [1.029–1.112]) and longer disease duration (OR = 1.004, 95% CI [1.000–1.007]) were independently associated with malignancy in male Chinese patients with RA, while older age (OR = 1.041, 95% CI [1.021–1.061]), positive RF (OR = 2.047, 95% CI [1.032–4.060]) and positive ACPA (OR = 2.677, 95% CI [1.312–5.463]) were associated with malignancy in female Chinese patients with RA.

Table 3 Multivariate logistic regression analysis of the associated factors with malignancy in Chinese patients and the US patients with RA*.

Characteristics	All RA	Male RA	Female RA	
OR (95% CI)	P	OR (95% CI)	P	OR (95% CI)	P	
Chinese RA patients							
Female	0.899 [0.525–1.540]	0.70	–	–	–	–	
Age	1.050 [1.032–1.068]	<0.001	1.070 [1.029–1.112]	<0.001	1.041 [1.021–1.061]	<0.001	
RA disease duration	1.001 [0.999–1.003]	0.19	1.004 [1.000–1.007]	0.04	1.001 [0.998–1.003]	0.63	
Active smoking	1.409 [0.907–2.190]	0.13	1.329 [0.627–2.817]	0.46	1.443 [0.622–3.347]	0.39	
Positive RF	1.693 [0.947–3.024]	0.08	0.800 [0.251–2.553]	0.71	2.047 [1.032–4.060]	0.04	
Positive ACPA	2.752 [1.492–5.076]	0.001	2.940 [0.991–8.721]	0.052	2.677 [1.312–5.463]	0.01	
CRP	1.001 [0.996–1.007]	0.66	0.994 [0.983–1.006]	0.32	1.005 [0.999–1.012]	0.12	
US RA patients							
Female	1.395 [1.113–1.749]	0.004	–	–	–	–	
Age	1.033 [1.024–1.042]	<0.001	1.070 [1.053–1.088]	<0.001	1.018 [1.008–1.028]	<0.001	
Active smoking	1.580 [1.260–1.983]	<0.001	1.348 [0.907–2.004]	0.14	1.627 [1.239–2.137]	<0.001	
Dyslipidemia	1.086 [0.832–1.419]	0.54	1.057 [0.696–1.606]	0.80	1.153 [0.811–1.641]	0.43	
Cardiovascular diseases	1.523 [1.213–1.911]	<0.001	1.214 [0.846–1.743]	0.29	1.975 [1.358–2.873]	<0.001	
Notes:

RA, rheumatoid arthritis; RF, rheumatoid factor; ACPA, anti-cyclic citrullinated peptide antibody; CRP, C-reactive protein; CDAI, Clinical Disease Activity Index; HAQ-DI, The Stanford health assessment questionnaire disability index.

* These variables which P < 0.1 in Table 2 (except for previous medication) were included in multivariate logistic regression analysis.

Differently, multivariate logistic regression analysis showed that female (OR = 1.395, 95% CI [1.113–1.749]), older age (OR = 1.033, 95% CI [1.024–1.042]), active smoking (OR = 1.580, 95% CI [1.260–1.983]) and cardiovascular diseases (OR = 1.523, 95% CI [1.213–1.911]) were independently associated with malignancy in the US patients with RA. Gender stratification showed that older age (OR = 1.070, 95% CI [1.053–1.088]) was independently associated with malignancy in male US patients with RA, while older age (OR = 1.018, 95% CI [1.008–1.028]), active smoking (OR = 1.627, 95% CI [1.239–2.137]) and cardiovascular diseases (OR = 1.975, 95% CI [1.358–2.873]) were associated with malignancy in female US patients with RA.

In the overall cohort, the result showed that the risk of malignancy prevalence in Chinese RA patients was lower than that in US RA patients (OR = 0.615, 95% CI [0.438–0.862]), especially in females (OR = 0.489, 95% CI [0.320–0.748], both P < 0.05, Table S2).

Discussion

This is the first population-based, cross-sectional study to assess the prevalence, subtypes and their associated factors of malignancy in a relatively large sample of patients with RA in China and the US. We found that Chinese patients with RA, especially females, had a relatively lower prevalence of malignancy than US counterparts. The peak interval of malignancy diagnosis was 3 years after RA onset in Chinese patients with RA. The malignancy showed a male predominance in China and female predominance in America. A comprehensive depiction of the atlas of malignancy prevalence, distribution and time interval in patients with RA will help in the prevention, early detection and proper management of cancer in patients with RA.

The malignancy burden in RA demonstrated considerable variability across countries, which exhibited a higher burden of malignancy in the developed countries in comparison to the developing countries (Dougados et al., 2014; Jin et al., 2015, 2017; Choi et al., 2018). In the COMORA study of 4,586 patients recruited in 17 participating countries, a history of any solid malignancies (excluding basal cell carcinoma) was found in 4.5% (95% CI [3.9–5.2%]) of patients with RA and ranged from a low of 0.3% in Egypt to a high of 12.5% in the US (Dougados et al., 2014). The prevalence of any malignancies was 5.0% in 1,050 RA patients from 11 Korean centers (Choi et al., 2018). Similarly, data from RA patients enrolled in the Chinese Registry of rhEumatoiD arthrITis (CREDIT) from November 2016 to August 2017 demonstrated that the prevalence of malignancy in a total of 13,210 RA patients was 0.6% (95% CI [0.5–0.7%]) (Jin et al., 2017). However, our study was the first comparative study to show that Chinese RA patients had a lower prevalence of overall malignancy than their gender- and age-matched US counterparts (6.2% vs. 12.6%). There were many confounding factors that may affect the risk of malignancy, Askling et al. (2016) found that there were significant variations in the standardized incidence rates of skin cancer in different countries (the US had a rate of 0.50 per 100 person-years, Sweden 0.17, the UK 0.25, Japan 0.01, and CORRONA International cohort 0.11) and hypothesized this was influenced by a range of factors, such as ethnic background, environmental exposures (including ultraviolet radiation), and regional variations in the diagnosis and reporting practices of non-melanoma skin cancer. On the other hand, smoking is a high-risk factor for malignancy worldwide, attributed to the occurrence of about 20 malignancies (Sasco, Secretan & Straif, 2004; Klebe et al., 2019). In our study, although the prevalence of smoking among RA patients in China is significantly lower than among RA patients in the US (16.1% vs. 58.2%), more RA patients with malignancy were active smokers in both Chinese and US patients. Multiple logistic regression analysis showed smoking was not associated with malignancy in Chinese RA patients, while active smoking (OR = 1.627) was associated with malignancy only in female US patients with RA. which might be one of the important factors related to more malignancy in the US patients.

Our comparative study revealed a huge variation in malignancies distribution between patients of China and the US with RA. Breast cancer, lung cancer, colon cancer, thyroid cancer and bladder cancer were more common in Chinese RA patients (65.4% of all cancer cases), while breast cancer, cervix cancer, non-melanoma skin cancer, uterus cancer and colon cancer were the most common cancer type in the US RA patients (65.6% of all cancer cases). Our study figures out gender discrepancies in cancer risk observed in RA. It is intriguing to observe a male predilection for RA with malignancy in China, while this finding is in contrast with those observed in the US. Previous studies had suggested that these differences may be due to differences in the environment, occupation and sex hormones between men and women. Recently, Li et al. (2018) discovered large differences in mutation density and sex biases in the frequency of mutation of specific genes; these differences may be associated with sex biases in DNA mismatch repair genes or microsatellite instability. Sex-biased genes include well-known drivers of cancer such as b-catenin and BAP1 (Li et al., 2018). In addition, the female genital organs (breast, cervix, and uteri) were the most common sites of cancer affecting female patients with RA in the US. This finding is in contrast with those observed in female patients with RA in China, the latter had a higher incidence of developing breast cancer, colon cancer, thyroid cancer, lung cancer and bladder cancer. Lung cancer is the most common for male patients with RA in China, followed by colon cancer and liver cancer, but the top three types of malignancy among male patients with RA in the US were non-melanoma skin cancer, prostate cancer and colon cancer. Moreover, the prevalence of nasopharyngeal cancer, stomach cancer, and liver cancer in patients with RA was higher in China than in the US, which may be related to infection. Nasopharyngeal cancer is a common malignancy in southern China and Singapore. It is consistently associated with genetic susceptibility and Epstein-Barr virus infection (Chen et al., 2019; Farrell, 2019). Infection with Helicobacter pylori (H. pylori) is the strongest risk factor for stomach cancer. The prevalence of H. pylori in China was approximately 20% higher than that in the US (Hooi et al., 2017). Hepatitis B virus (HBV) infection is a major cause of chronic hepatitis, liver cirrhosis, and hepatocellular carcinoma in humans. Our previous study showed that the prevalence of HBV infection in our cohort of patients with RA was as high as 10.1% (Zheng et al., 2024). The risk of developing different types of cancer varies among patients with RA by region, gender, age, and infection status, so more aggressive monitoring of cancer in patients with RA during their lifetime management is warranted and screening for cancer should be individualized.

Prior studies have demonstrated that patients with RA appeared to have a higher risk of malignancy compared with the general population (Simon et al., 2015; Beydon et al., 2023; Wang et al., 2023). However, the mechanism remains an ongoing debate (Baecklund et al., 2006; Chatzidionysiou et al., 2022). Recently, Wang et al. (2023) reported that RA patients were 1.69 to 2.08 times more likely than those without RA to develop lymphoma or lung cancer within 1 year of RA diagnosis, however, commonly used RA treatments were unlikely to increase cancer risk within 1 year. Chatzidionysiou et al. (2022) furtherly reported that in a Swedish population-based cohort study with a mean follow-up of 7.3 years, seropositivity (RF and/or ACPA positive) were associated with increasing 2–6 times higher incidence of lung cancer, even when adjusted for smoking. Similar results were also shown in our study, positive ACPA (OR = 2.752) was associated with a higher likelihood of malignancy in Chinese patients with RA. Moreover, positive RF (2.0-fold) and positive ACPA (2.7-fold) were associated with higher odds of malignancy in female Chinese patients with RA, respectively. There are several potential explanations for this association. There is increasing evidence that mucosal sites, and in particular, the lungs, might be the site of generation of RA-associated autoimmunity. RA-associated antibody enrichment and immune activation may promote inflammation. This early inflammatory process could potentially activate carcinogenic mechanisms. On the other hand, the presence of autoantibodies like RF and ACPA can lead to immune dysregulation, which may impair the body’s ability to surveil and eliminate neoplastic cells (Fragoulis & Chatzidionysiou, 2020). Moreover, the peak interval of malignancy diagnosis was 3 years after RA onset. Those imply a need for increased awareness for regular cancer screening in subset of seropositive RA, especially in the near-term following RA onset.

Moreover, previous studies reported that the use of tumor necrosis factor (TNF) or interleukin (IL)-6 inhibitors was not associated with an increased risk of cancer development in patients with RA, even TNF inhibitor was associated with a lower cancer incidence (Choi et al., 2022; Huss et al., 2022). Furthermore, anti-TNFα therapy after cancer diagnosis did not influence recurrent or new primary cancer development (Waljee et al., 2020). While tofacitinib and abatacept were controversially associated with a slight increased risk of cancer compared with other bDMARDs or bDMARDs-naїve (Montastruc et al., 2019; Simon et al., 2019; Curtis et al., 2023; Sepriano et al., 2023; Xie et al., 2020). Thus, 2022 EULAR points to consider that JAK inhibitors and abatacept may be used with caution in patients with a history of cancer, anti-cytokine bDMARDs may be preferred in patients with a history of solid cancer, and B cell depleting therapy may be preferred in patients with a history of lymphoma (Sebbag et al., 2023). The association between glucocorticoid therapy and the risk of malignancy remains a subject of considerable debate. There were two studies exploring the association between steroids and lymphomas in RA patients, one indicated no increase of risk (Engels et al., 2005) but the other indicated an increased risk (Smedby et al., 2006). Both studies didn’t take into account RA disease severity, treatment or accumulated steroid exposure/route of administration. One more recent study encompassed a cohort of 74,651 RA patients, among whom 378 were diagnosed with lymphoma, indicated that a total duration of oral glucocorticoid treatment of <2 years was not associated with lymphoma risk (OR = 0.87, 95% CI [0.51–1.5]), whereas total steroid treatment >2 years was associated with a lower lymphoma risk (OR = 0.43, 95% CI [0.26–0.72]). These results indicated that total duration of time receiving steroid treatment is important (Hellgren et al., 2010). Chronic inflammation is a well-known facilitator of malignancy, thus the main mechanism by which steroid treatment reduces the risk of lymphoma in RA patients may be related to its robust anti-inflammatory effect. In addition, steroids are known to induce apoptosis in lymphatic cells, which may also contribute to the effects observed with prolonged steroid treatments employed in RA, potentially modulating emerging clonal B-cell populations (Mizushima et al., 2023). In our study, we found that Chinese RA patients with malignancy were more likely to have previous uses of csDMARDs (83.1% vs. 69.4%) and glucocorticoids (61.9% vs. 48.9%), but less uses of bDMARDs (1.7% vs. 7.2%) than those without malignancy, while there was no difference of medication treatment in US RA patients with or without malignancy. However, given the nature of cross-sectional investigation, the causality between RA treatment and malignancy risk cannot be determined in this study, and further prospective cohort studies are needed to validate the relationship between RA treatment and malignancy risk.

There were several limitations in this study. Firstly, the data were cross-sectionally collected, and the Chinese RA patients were recruited from two centers in southern China, which may not fully represent the entire Chinese RA population, leading to potential selection bias. Furthermore, the lack of controls subject in both China and the United States made it impossible to compare the relative risk for malignancy of RA. Secondly, detection of RA and malignancy diagnosis were derived from questionnaires and self-reports in NHANES database, but not from a RA-specific registry, which might lead to diagnostic bias and lack of detailed RA-related clinical data, and potential screening bias of malignancy events should also be considered. However, though the diagnosis of malignancy in NHANES was based on interviews and relied on self-reports, the prevalence of malignancy in RA patients in NHANES database (14.5%) was similar with RA patients in the United States reported by COMORA study (12.5%), in which the information of malignancy was being collected more strictly, including gathering by a study investigator during a face-to-face interview at a dedicated study visit, through review of the medical record and relevant screening, including laboratory and imaging examinations (Dougados et al., 2014). This data provides further support for our conclusion of lower prevalence of malignancy in Chinese RA patients than that in American RA patients, even though major differences in cancer diagnosis ascertainment between the NHANES and our study. Thirdly, the calendar years of included RA patients were different from China and the US, because time trends of some malignancy incidence changed significantly in the past twenty years, future comparative studies are needed to balance this potential bias. Fourthly, malignancy in RA patients might impact their therapeutic schedule, prognosis and outcomes (Zhang et al., 2020; Fujita et al., 2024; Pundole et al., 2020). However, we didn’t compare these data in this cross-sectional study. Future follow-up research is needed to answer whether different spectrum of malignancy will cause different outcome, in particular, whether the higher rate of skin cancer in America RA patients means a better overall prognosis than those with more serious pathology pertaining to breast and lung cancer in Chinese RA patients.

In conclusion, the prevalence of overall malignancy in China was two times lower than in the US patients with RA. It is intriguing to observe a male predilection for RA with malignancy in China, while this finding is in contrast with those observed in the US. Considering the increased cancer risk in the long-term management of patients with RA, increasing awareness and development of individualized cancer screening strategies for patients with RA are warranted.

Supplemental Information

Supplemental Information 1 Chinese patients with RA.

The raw data shows that the prevalence of malignancy in Chinese patients with RA is nearly twice lower than that in the US (6.2% vs. 12.6%).

Supplemental Information 2 US NHANES database.

Supplemental Information 3 Flow chart of patient enrollment.

RA, rheumatoid arthritis.

Supplemental Information 4 The clinical characteristics of Chinese patients with malignancy after RA onset.

RA, rheumatoid arthritis; RF, rheumatoid factor; ACPA, anti-cyclic citrullinated peptide antibody; 28TJC, 28-joint tender joint counts; 28SJC, 28-joint swollen joint counts; PtGA, patient global assessment of disease activity; PrGA, provider global assessment of disease activity; Pain VAS, pain visual analog scale; ESR, erythrocyte sedimentation rate; CRP, C-reactive protein; CDAI, Clinical Disease Activity Index; HAQ-DI, health assessment questionnaire disability index.

Supplemental Information 5 Association of different country with malignancy in the overall cohort in an adjusted model.

Model 1: Unadjusted; Model 2: Adjusted for age, gender (male or female); Model 3: Adjusted for model 2 covariates plus BMI, active smoking (yes or no), CRP, hypertension (yes or no), diabetes (yes or no), dyslipidemia (yes or no), cardiovascular diseases (yes or no), enrollment year interval and previous medication.

Additional Information and Declarations

Competing Interests

Author Contributions

Human Ethics

Data Availability

The authors declare that they have no competing interests.

Zhi-Ming Ouyang conceived and designed the experiments, performed the experiments, analyzed the data, prepared figures and/or tables, authored or reviewed drafts of the article, and approved the final draft.

Yao-Wei Zou conceived and designed the experiments, performed the experiments, analyzed the data, prepared figures and/or tables, authored or reviewed drafts of the article, and approved the final draft.

Jie Pan conceived and designed the experiments, performed the experiments, analyzed the data, prepared figures and/or tables, authored or reviewed drafts of the article, and approved the final draft.

Ye Lu conceived and designed the experiments, performed the experiments, analyzed the data, prepared figures and/or tables, authored or reviewed drafts of the article, and approved the final draft.

Ying Yang conceived and designed the experiments, performed the experiments, analyzed the data, prepared figures and/or tables, authored or reviewed drafts of the article, and approved the final draft.

Qian-Hua Li conceived and designed the experiments, performed the experiments, analyzed the data, prepared figures and/or tables, authored or reviewed drafts of the article, and approved the final draft.

Jian-Da Ma conceived and designed the experiments, performed the experiments, analyzed the data, prepared figures and/or tables, authored or reviewed drafts of the article, and approved the final draft.

Pei-Wen Jia conceived and designed the experiments, performed the experiments, analyzed the data, prepared figures and/or tables, authored or reviewed drafts of the article, and approved the final draft.

Tao Wu conceived and designed the experiments, performed the experiments, analyzed the data, prepared figures and/or tables, authored or reviewed drafts of the article, and approved the final draft.

Yu-Ting Fan conceived and designed the experiments, performed the experiments, analyzed the data, prepared figures and/or tables, authored or reviewed drafts of the article, and approved the final draft.

Jian-Zi Lin conceived and designed the experiments, performed the experiments, analyzed the data, prepared figures and/or tables, authored or reviewed drafts of the article, and approved the final draft.

Xiu-Ning Wei conceived and designed the experiments, performed the experiments, analyzed the data, prepared figures and/or tables, authored or reviewed drafts of the article, and approved the final draft.

Kui-Min Yang conceived and designed the experiments, performed the experiments, analyzed the data, prepared figures and/or tables, authored or reviewed drafts of the article, and approved the final draft.

Yun Su conceived and designed the experiments, performed the experiments, analyzed the data, prepared figures and/or tables, authored or reviewed drafts of the article, and approved the final draft.

Lie Dai conceived and designed the experiments, performed the experiments, analyzed the data, prepared figures and/or tables, authored or reviewed drafts of the article, and approved the final draft.

The following information was supplied relating to ethical approvals (i.e., approving body and any reference numbers):

The Ethics Committee of Sun Yat-sen Memorial Hospital approved this study (SYSEC-2009-06). All patients consented to participate and signed informed consent.

The following information was supplied regarding data availability:

The raw measurements are available in the Supplementary Files.

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
