# Peer review of "Different prevalence and spectrum of malignancy between Chinese patients and American patients with rheumatoid arthritis: a comparative study"

_PeerJ, doi:10.7717/peerj.18650_

## Round 0.1 · original submission · Major Revisions

The authors are requested to carefully revise the manuscript and answer the questions raised by the expert reviewers.

Reviewer 1 ·

Basic reporting

The description of the purpose of the study is clear. It is great to see study with large population size, and authors are able to find comparison group. A few comments here:
1. Line 54, 95% CI stands for 95% confidence interval, not confidence index.
2. Line 142-145, looks like Chinese patients and American patients are 1:1 mapping? Since they are stratified by age first, and matching individual patient within each gender stratum based on age. Then why there are more US patients in the study?
3. Line 149, what are the benefits of using sampling weights?
4. Line 173, why mean age contains 95% CI?

Experimental design

I think the design of the study is in good concept, and the statistical analyses are rigorous. However, I have a few major concerns:
1. In multiple places, it was mentioned that the matching is "individual matching", then it's a bit confusing why the two population have unequal numbers?
2. Authors did not explain why the weights in Chinese patients with RA were set to 1?
3. Did authors ever considered selection bias issue of the Chinese patient group? The prevalence of malignancy were much lower, and the number of this cohort is 118, which significantly drop the research population size.

Validity of the findings

Since the Chinese patients data were patients enrolled from 2001 to 2024, which across long period of time, and this potentially cause trouble for retrospective study like this. Within 20 years, people's lifestyle were changed drastically and the reason for causing malignancy among RA patients can have many confounding factors. Same for US patients, selected from 1999-2018. Therefore, the conclusion that US and Chinese patients have drastic difference in the malignancy may not be a solid conclusion since there are many confounding factors, and only addressing age/gender may not be sufficient. Malignancy and RA have many reasons, and they could be correlated by themselves. Authors should consider more influence before making conclusions.

·

Basic reporting

This paper compares and contrasts the prevalence of malignancy in Chinese patients with rheumatoid arthritis with American patients with rheumatoid arthritis. It reports a male predominance of malignancy in China and female predominance in America. Hence, although the overall prevalence of malignancy among American patients with rheumatoid arthritis was twice that seen in Chinese patients, this difference was only statistically significant for females. Subtypes of malignancy also differed between the 2 groups with breast cancer twice as common among Americans with rheumatoid arthritis than in Chinese with rheumatoid disease. Older age carried a lower odds ratio for malignancy than positive serology, smoking and cardiovascular disease (the latter in America).

The English is generally good but there are 2 or 3 sentences which do not make sense as a result of the English requiring adjustment. The references, introduction, and discussion were all context Jewell and appropriate. The article is structured professionally and raw data are shared in a transparent way.

Experimental design

The design was based on comparing two large cohorts of around 2000 patients with rheumatoid arthritis within both China and America. However ascertainment of the diagnosis of cancer appeared much more rigorous in China where medical records were accessed and pathology obtained to confirm the diagnosis. In America diagnosis was based on interviews and relied on self reports. As far as I could tell, there was no requirement for access to medical records and certainly none to pathology in American patients. This alone is likely to induce a significant bias and could at least in theory, explain the whole difference. Although this is admitted within the Discussion, I think it is difficult to ascertain the contribution that this bias makes to the results but it is very important to acknowledge this may be a potentially major flaw.

Furthermore, there were no controls in either country to determine the rates of malignancy in age and sex matched individuals without rheumatoid arthritis. Although the authors make detailed reference to previous studies showing more malignancy in patients with rheumatoid arthritis, it cannot be safely assumed that this would necessarily also apply to this study, especially given the differences in the mechanisms for ascertainment of cancer diagnosis between the two nations.

Validity of the findings

Although the diverse ethnicity within Americans is acknowledged, discussion as to the influence this may have on the age of onset and types of malignancy within the USA is missing. This may be particularly pertinent with reference to breast cancer which is shown to be twice as prevalent in America as in China (see text, results, tables 2A and 2B). The rates of skin cancer are also much increased in America and reasons for this may include genetic variation as well as sun exposure. In China, the highest prevalence of cancer related to lungs but it is unclear as to how much smoking contributed towards this. Was smoking controlled for when comparing the prevalence of lung cancers between the 2 countries?

Of the 118 Chinese people with rheumatoid arthritis and malignancy, 37% were diagnosed with malignancy before the onset of rheumatoid arthritis. Can we also be sure that this was therefore related to rheumatoid arthritis? How these data compare with the American data? I could not find an answer to this. Within the Discussion (beginning of paragraph 3), it is stated that breast cancer is, along with genital cancer, more common in China. This is at odds with the results which indicate that the prevalence of breast cancer is twice as great in America.

Additional comments

1 I would suggest that the title be altered to emphasise that the differences in rates for malignancy between the 2 countries apply only to females.
2 Although paragraph 5 in the discussion make some reference to treatment, it would be helpful to know whether there was any correlation between different forms of treatment and different types of cancer in both countries.
3 Is there a difference in prognosis and outcomes for malignancy between China and America? In particular, does the higher rate of skin cancer in America mean that these patients carry a better overall prognosis than those with more serious pathology pertaining to breast and lung in China? This is important as the headline title suggests that cancer is more common in Americans but this may not be applied to outcomes which would be an important clarification to make once the answer is established.

·

Basic reporting

vide infra

Experimental design

vide infra

Validity of the findings

vide infra

Additional comments

1. Line 90: “The patients with RA were diagnosed according to the 1987 American College of Rheumatology (ACR) [19] or 2010 ACR/ European League Against Rheumatism (EULAR) criteria [20]” – please note that these are classification criteria, not diagnostic criteria. RA is diagnosed clinically. A patient can have RA without fulfilling these criteria, and likewise a patient fulfilling these criteria cand have another form of arthritis. So, please rephrase to say that your patients were diagnosed with RA and these diagnoses also fulfilled the cited criteria.
2. Line 86: you recruited patients from 2 Chinese centers. Are these representative for the Chinese RA population? Please clarify in the Methods and/or Discussion.
3. Line 86: you recruited patients from 2 Chinese centers. The study population is highly variable, so please briefly state in the Methods and/or Discussion some elements that contribute to this variability: real life patients, multiple health care professionals who diagnosed them, multiple health care professionals who evaluated their RA activity.
4. Line 86: you recruited patients from 2 Chinese centers. You offer no information on treatment. RA treatment (NSAIDs, glucocorticoids, csDMARDs, bDMARDs, tsDMARDs) can impact cancer risk. Either report this information if you have it or Discuss its lack.
5. Line 159: “statistical software packages IBM SPSS 25.0” – please note that the software is not properly cited. Please see: https://www.ibm.com/support/pages/how-cite-ibm-spss-statistics-or-earlier-versions-spss
6. Table 1: please add normal range for CRP in the footnotes.

---

## Round 0.2 · Minor Revisions

The authors are requested to carefully revise the manuscript and answer the final questions raised by the reviewers.

Reviewer 1 ·

Basic reporting

No comments.

Experimental design

No comments.

Validity of the findings

No comments.

·

Basic reporting

The use of English language is good, and the References are expanded and appropriate. The background discussion has been expanded well. No concerns regarding Tables / Figures etc.

Experimental design

The authors have addressed most of my concerns in these areas. Significant limitations are admitted and these include (1) major differences in cancer diagnosis ascertainment between the US and China, (2) the lack of control subjects in both countries (making it impossible to compare the RR for malignancy of RA), and (3) the absence of any follow-up or therapeutic data making an assessment of prognosis impossible. The greater number of skin cancers in the USA account for a significant part of the twofold increased cancer rate in the US over China, and this is likely to be reflected in a better prognosis.

Hence, although the research question is addressed and answered, significant nuances persist but are now more openly admitted and discussed. Methods are transparent enough to facilitate study replication.

Validity of the findings

The findings are valid enough and the limitation of the interpretation of the findings is discussed in the previous paragraph.

It would be interesting to hear the authors explanation of why they found an association between positive RF / ACPA and malignancy, and also the reason why there was an association between cancer and steroid therapy in Chinese patients.

The conclusions are now more discursive and reflective which is encouraging.

Additional comments

No further comments

·

Basic reporting

no comment

Experimental design

no comment

Validity of the findings

no comment

Additional comments

the authors have properly addressed the raised issues

---

## Round 0.3 · accepted · Accept

After revisions, all reviewers agreed to publish the manuscript. I also reviewed the manuscript and found no obvious risks to publication. Therefore, I also approved the publication of this manuscript.

·

Basic reporting

I am happy with this revision as it addresses my concerns adequately in the Discussion section

Experimental design

Satisfactory

Validity of the findings

Acceptable

Additional comments

None